# A Single Bout of Physical Exercise Mimicking a Motor Seizure Increases Serum MMP-9, but Not S100B

**DOI:** 10.3390/ijms26051906

**Published:** 2025-02-23

**Authors:** Jan Karol Sielczak, Maciej Krawczyk, Agnieszka Cudna, Iwona Kurkowska-Jastrzębska

**Affiliations:** Second Department of Neurology, Institute of Psychiatry and Neurology, 02-957 Warsaw, Poland; jsielczak@ipin.edu.pl (J.K.S.); krawczyk@ipin.edu.pl (M.K.); acudna@ipin.edu.pl (A.C.)

**Keywords:** epilepsy, seizures, metalloproteinase, MMP-9, S100B, physical effort, biomarkers

## Abstract

MMP-9 and S100B, the proteins involved in blood-brain barrier integrity, are widely studied as biomarkers in many diseases, including epilepsy. They are elevated in epilepsy patients both interictally and following motor seizures. To determine whether motor activity influences their serum concentrations, we investigated the effects of brief, seizure-like physical exercise on serum MMP-9 and S100B levels in healthy individuals. Participants performed two different 5-min exercise protocols mimicking the motor activity of bilateral tonic–clonic seizures, one of the sets of exercises that contribute to to muscle failure. Serum samples were collected before exercise, 3 h after exercise, and the next day (time points 0, 3 h, and 24 h). Our results demonstrated that both sets of motor exercises led to a similar increase in MMP-9 levels, while neither affected S100B levels. No significant differences in MMP-9 levels were observed due to muscle failure. We suggest that the increase in MMP-9 seen after seizures is induced partially by peripheral mechanisms, such as muscle contraction. S100B appears to be a promising biomarker in epilepsy, as it is not induced by physical activity but does increase following seizures. Further research is needed to fully elucidate the mechanisms underlying biomarker release in epilepsy and to determine the specific contributions of muscle contractions versus other seizure-related processes.

## 1. Introduction

Epilepsy is a common disease that affects people of all ages and genders. Patients struggling with epilepsy are at risk of seizures, which often occur for no apparent reason. The cause of epilepsy might be genetic, metabolic (with a genetic basis or acquired deficiencies), or as a result of brain disease or damage, such as infections, immune reactions, and structural abnormalities (stroke, trauma, etc.), however, in about 50% of patients the cause remains unknown. Epilepsy is manifested by seizures of various motor and non-motor neurological signs and loss of consciousness [1]. Unfortunately, despite the intensive development of pharmacological treatment, many patients fail to completely reduce seizures; it is estimated that this applies to approximately 30% of patients [2].

There is growing interest in searching for serum biomarkers of epileptic seizures that could help in the correct diagnosis of seizures (epileptic vs. non-epileptic), assessment of seizure risk, and prediction of treatment outcome. MMP-9 and S100B, the proteins involved in blood-brain barrier integrity, are particularly interesting because they are increased in epilepsy patients in the interictal period [3] and are triggered by the seizures themselves [4,5].

Recently, several studies have suggested the involvement of the blood-brain barrier (BBB) in the pathogenesis of both epilepsy and seizure induction. An increase in the permeability of the BBB seems to be the cause of increased neuronal excitability. On the other hand, seizures, and even epileptic discharges, increase BBB permeability and activation of endothelial cells. This was shown in experimental studies [6] as well as in brain samples taken from epileptic foci in human [7]. The serum biomarkers of BBB were shown to be increased in epileptic patients after seizures [8] and in the interictal period when compared to the age- and sex-matched control group [3]. These observations indicate chronic BBB disruption in epileptic patients probably due to chronic inflammation, and BBB involvement in seizures inducement. Additionally, higher levels of MMP-2 and CCL-2 in the interictal period were associated with a lower risk of seizures, whereas a higher level of MMP-9 was associated with a higher risk of seizures in a 1–6-month period [9].

However, looking for reliable BBB biomarkers has limitations since most of them are not specific for the brain and might originate from peripheral vasculature and peripheral tissues. Many of them, for example, MMP-9, P-selectin, and ICAM-1 participate in brain inflammatory responses and may be involved in many brain pathologies. Muscle contractions during focal motor or tonic–clonic seizures can increase serum biomarkers due to muscle vessel dilation and injury. For example, MMP-9 concentrations are known to change after exercise, suggesting that muscle contractions during seizures could partially contribute to the observed increase in these biomarkers following seizures [10].

MMP-9 is a metalloproteinase highly expressed in the adult brain and has been shown to increase in many brain pathologies. The serum MMP-9 level depends mainly on leukocyte and endothelial secretion, and augments significantly in many conditions (such as trauma or stroke) [11]. MMP-9 contributes to a wide variety of brain disorders, including epilepsy, schizophrenia, autism spectrum disorders, brain damage, stroke, neurodegeneration, pain, and brain tumors [12]. It activates various cytokines and chemokines, participates in breaking the blood-brain barrier, and facilitates the passage of leukocytes across the BBB into the brain parenchyma.

The S100B protein is physiologically produced and released predominantly by astrocytes in the central nervous system [13]. It plays a role in many cellular processes, including cell proliferation, migration, apoptosis, and differentiation. S100B has been shown to increase in cerebrospinal fluid (CSF) and serum after various neurological diseases (for example, trauma, stroke, inflammation, bacterial and viral meningitis) and it has been postulated that S100B could serve as a serum marker for brain damage.

Previous studies have demonstrated that tonic–clonic seizures in individuals with epilepsy result in elevated serum concentrations of MMP-9 and S100B. Peak levels are typically observed 3–6 h after seizure onset, with elevations persisting for up to 24 h [3,4,5,8,9,10]. To investigate whether muscle contractions contribute to the release of these biomarkers, we conducted an experiment with healthy participants. Participants performed exercises designed to mimic the physical exertion of bilateral tonic–clonic seizures. This controlled exercise paradigm allowed us to examine the impact of muscle contractions on biomarker levels, independent of seizure activity. The exercise protocol was carefully designed to minimize potential biases, and the blood sampling schedule mirrored that used in our previous studies with epilepsy patients. We observed that two distinct sets of short exercises, one engaging multiple muscle groups and the other targeting specific muscles, led to increased serum MMP-9 concentrations. However, the pattern of MMP-9 elevation differed from that observed following seizures. Notably, the S100B levels remained unchanged after exercise.

## 2. Results

Sixteen participants (4 men and 12 women) took part in the study. The mean age was 32.37 ± 9.23 years, and the range of age was from 22 to 55 years old. Both sets of exercises (E1 and E2) were performed for all the groups of participants approximately 4 months apart from each other. The basic levels of MMP-9 were 583.45 ± 385.26 ng/mL and 603.33 ± 478 ng/mL before the first and second physical efforts, respectively (no statistical difference was noted). The basic levels of S100B were 32.28 ± 5.35 and 38.06 ± 9.05 pg/mL before the second effort (E2) and both values were slightly different (*p* < 0.025).

The level of MMP-9 increased significantly after the E1 and E2 sets of exercises and was higher 3 and 24 h after the efforts (Table 1, Figure 1). The S100B levels did not increase after the efforts regardless of the type of exercise. The levels of MMP-9 during both E1 and E2 physical efforts did not differ at the same time points—the increases after 3 h and 24 h following the exercises were comparable for both exercise types (p2 from Table 1 describes the comparison of E1 and E2 biomarkers levels at the same time points). The levels of S100B were slightly different only on the basic level (time points—0 h) between E1 and E2 measurements.

The groups were not numerous, and no correlation of sex or age was noted. The basic levels of MMP-9 were strongly correlated with a higher level 3 and 24 h after the efforts in both E1 an E2 exercise sets. The S100B level did not correlate with MMP-9 at any time point.

## 3. Discussion

Our study demonstrated that brief but very intense anaerobic exercise increased serum MMP-9 levels, which persisted for 24 h. In contrast, serum S100B levels remained unchanged. These findings indicate that serum MMP-9 is sensitive to physical activity, which should be considered when interpreting MMP-9 levels in pathological conditions, particularly those associated with exertion, such as motor seizures.

Current evidence suggests that intense exercise can influence serum MMP-9 levels, although some studies report no or only a slight effect [10,14,15]. Factors such as the type and duration of exercise, muscle and tissue damage, and underlying health conditions may contribute to these varied outcomes [16,17]. Short-term exercise tends to elevate MMP-9 levels, whereas prolonged training may decrease baseline serum MMP-9, as observed in both healthy volunteers and athletes [10,14,15,18]. In individuals with chronic conditions associated with elevated MMP-9, regular physical activity may help reduce these levels [16,17,19]. Our study included exercises of different characteristics, including one predominantly anaerobic, which was unique not only in its metabolic profile but also in its ability to mimic tonic–clonic seizures. Short, intense anaerobic exercise, as well as even more intense forms of daily physical activity, can influence biomarker levels, which should be considered when interpreting them in the context of epilepsy. These findings highlight the importance of accounting for physical exertion when analyzing biomarkers, as even brief and intense activities can impact their levels.

In epilepsy, MMP-9 levels are higher compared to the age- and sex-matched control group [3] and increase after tonic–clonic seizures reaching the level by about 2 times higher than the age-matched control [4,5]. Epileptic seizures are thought to induce inflammation and blood-brain barrier leakage which may be the cause of the increase in MMP-9 and S100B, along with other biomarkers in the serum [10]. However, the elevation of some biomarkers may be caused by muscle contraction, increased blood flow, or increased adrenergic stimulation during seizures. Our study aimed to develop exercises that mimic bilateral tonic–clonic seizures and confirm that these exercises are responsible for an increase in MMP-9, but not in S100B.

When comparing our results to previous studies on epilepsy, the increases in MMP-9 and S100B levels after seizures appear to be higher and more prolonged. In those studies, MMP-9 and S100B concentrations remained elevated for at least 24 h after seizures, with MMP-9 persisting at elevated levels for up to 72 h [4,5,8]. Nass et al. [5] reported a 50% increase in MMP-9 and an 80% increase in S100B following seizures. Similarly, Cudna et al. [4,8] found that MMP-9 and S100B concentrations were approximately 100% higher than control levels three hours post-seizure. However, because baseline biomarker levels for the epilepsy group were not reported in this study, direct comparisons with our findings are difficult. In our current study, we observed a 50% increase in MMP-9 concentration following an exercise session, with no corresponding rise in S100B. Our results suggest that physical activity induces a smaller and shorter increase in MMP-9 concentrations compared to seizures and does not elevate S100B levels. Further investigation into non-motor seizures could help clarify the extent to which serum MMP-9 increases are driven by muscle activity versus brain epileptic activity. Most studies on MMP-9 and S100B serum levels focus on generalized tonic–clonic seizures or febrile seizures, with limited data on non-motor seizures. Recent meta-analyses of serum MMP-9 concentrations have also reported consistent findings across studies [20].

The mechanism that causes the increase in MMP-9 concentration in serum after exercise could be related to several factors. Matrix metalloproteinase-9 (MMP-9) is an enzyme involved in the breakdown of extracellular matrix proteins, playing a role in tissue remodeling and repair. Physical exercises, especially intense or prolonged, can induce inflammation and tissue damage. MMP-9 is produced and released by various cells, including neutrophils and macrophages. Physical exercise can stimulate the mobilization and activation of these cells, leading to increased MMP-9 release. Reihmane et al. [21] showed that a short 2-min exercise induced leukocytosis and increased IL-6, myeloperoxidase (MPO) and MMP-9 concentrations. Since increased MPO (marker of neutrophils degranulation) correlated with an increase in IL-6 and MMP-9 concentrations, neutrophils could be the main source of these inflammatory proteins during maximal effort exercise. They also suggested that the increase in MMP-9 was not due to physiological damage to muscle fibers and connective tissue caused by physical exercise, as creatine kinase levels remained unchanged. Similarly, our study found no differences in MMP-9 levels based on the type of exercise or the presence of muscle tissue damage. Both exercise protocols resulted in comparable increases in MMP-9, despite one (E2) inducing muscle failure while the other (E1) did not. Other authors have also shown that MMP-9 levels increase after exercise, independent of the extent of tissue damage and the rate of recovery [22].

On the other hand, various types of systematic training reduce the MMP-9 level. It is very well known that regular physical exercise contributes to reducing inflammation through multiple pathways, including the reduction in adipose tissue, improved insulin sensitivity, increased production of anti-inflammatory cytokines, enhanced immune regulation, reduced oxidative stress, and regulation of stress hormones. This mechanism may lead to MMP-9 decrease after a long training both in healthy people and in patients with inflammatory states such as diabetes [16,17] or cancer [19], and may be responsible for the good effect of training on the outcome of these diseases.

S100B has also been studied as one of the markers during sports activity. The S100B protein is mainly related to damage to the central nervous system since it is expressed in astrocytes and certain neuronal populations (however, it is also expressed in many other cells, for example, Schwann cells, melanocytes, chondrocytes, adipocytes, skeletal myofibers, and lymphocytes) [23,24]. The transient increase in S100B was shown after intensive aerobic efforts such as a long-distance swimming race [25], high-intensity intermittent exercise or moderate continuous exercise on bicycle ergometer [26] or training on a treadmill [27]. The mechanism of S100B increase is not known and may include the release of S100B from cerebral vessels. Additionally, increased adrenergic stimulation during exercise may stimulate the adipose tissue to lipolysis and the release of S100B [28]. Similarly to MMP-9, the S100B protein level decreases after a prolonged, systematic training process [24].

In our study, the relatively short and highly anaerobic stimulation with strength-type exercise did not result in an increase in S100B. The characteristics of the exercise protocol may explain this result. Cardiorespiratory exercises may induce an increase in S100B, unlike the strength-based exercises used in our study. Additionally, the timing of blood collection could be a factor. In the studies mentioned above, blood was collected immediately after exercise (within minutes), and S100B levels might return to baseline very quickly [26,27]. In one study, blood taken one hour after exercise showed no significant differences [25].

Briefly elevated S100B levels following intense exercise do not indicate pathological brain damage or blood-brain barrier impairment, but rather a temporary physiological response to exercise stress. On the other hand, persistently elevated S100B levels may indicate brain pathology. Numerous studies have shown a correlation between S100B concentration and various types of concussion [29]. Chronically elevated MMP-9 and S100B in epilepsy patients above the levels observed in age-matched controls might therefore indicate neurodegenerative processes (such as in mesial temporal sclerosis) or epileptic discharge activity [3]. The concentration of certain biomarkers has been suggested as a potential predictor of epileptic seizures. Serum levels of MMP-2, MMP-9, and CCL-2 have been found to predict seizure frequency over a 12-month period. These biomarkers may reflect underlying processes such as blood-brain barrier activation, neuronal excitability, and inflammation, which are involved in seizure initiation [9].

The main limitation of our study is the relatively small sample size, although we examined the same group twice, which probably increases the importance of the investigation. We acknowledge that increasing the number of participants could enhance the statistical power of our findings. However, this study was conducted in a highly controlled environment, with healthy participants performing both sets of exercises. All exercise sessions were carefully supervised and standardized. This rigorous approach allowed us to explore the potential relationship between exercise and the levels of MMP-9 and S100B under consistent conditions, yielding valuable insights. Our study is the first to examine the impact of pure anaerobic exercise on changes in MMP-9 and S100B levels.

## 4. Materials and Methods

### 4.1. Participants

The study was approved by the Committee for Ethics in Human Research at the Institute of Psychiatry and Neurology (Warsaw, Poland, Decision No. 18/2021) and was performed in the 2nd Department of Neurology, Institute of Psychiatry and Neurology in Warsaw.

The participants were recruited among healthy volunteers from hospital staff. The participants were examined to exclude contraindications to intense physical exercise. Before participating in the study, each subject underwent a detailed medical interview and health assessment. The interview included questions regarding chronic medication use and the presence of cardiovascular, respiratory, metabolic, autoimmune, or other systemic diseases. Particular attention was paid to the level of physical activity—individuals regularly performing high-intensity resistance-endurance training were excluded from the study. Additionally, special consideration was given to questions regarding excessive musculoskeletal loading, which could alter biomarker concentrations and potentially interfere with study outcomes. All included participants reported that their physical activity did not exceed the level required for daily activities and occasional walks, and none engaged in systematic physical training. The study group consisted exclusively of clinical staff members, including physicians, physiotherapists, neuropsychologists, speech therapists, and occupational therapists.

The following inclusion criteria were used in the study group: age over 18 and signed written informed consent.

The exclusion criteria were as follows: obesity and metabolic syndrome, BMI above 29.9, any medication for the last 5 days, active infection, any cardiovascular and neurological disorder, immunosuppressive or immunomodulatory treatment in the last six months, surgery or significant trauma within the last two months, hepatic or renal insufficiency, pregnancy, clinical and laboratory symptoms of infection, muscle tear in the last four weeks, bone fractures, severe cardiorespiratory disease, active strength training, active delayed onset muscle soreness, or a high-intensity training session in the last week.

The first physical effort (E1) consisted of two phases designed to replicate the muscle contraction patterns observed during tonic–clonic seizures. In the first phase, participants assumed a classic forearm plank position and maintained it until complete muscular exhaustion, but for no longer than one minute. This phase aimed to induce global isometric muscle tension, particularly in the trunk, simulating the tonic phase of a seizure. A proper technique was required, but the primary focus was on achieving maximal overall isometric contraction. The second phase involved rapid tapping of the feet against the ground while maintaining a push-up position on extended arms for five minutes. Participants were instructed to perform the movement as quickly and as intensely as possible throughout the duration of the task. Emphasis was placed on tempo and movement intensity rather than the range of motion of the lower limbs. This phase was intended to model the clonic phase of a seizure, characterized by rhythmic, forceful contractions of the lower limb muscles. If a participant was unable to sustain the push-up position for the full five minutes, they were required to immediately switch to a back support position, with their hands placed behind them and hips elevated (bridge position), continuing the tapping exercise at the same intensity, duration, and rhythm as in the original position. The exercise primarily engaged trunk-stabilizing muscles working isometrically, especially during the plank phase. In the second phase, additional involvement of the lower limb muscles was observed, including the erector spinae, gluteus maximus and medius, quadriceps, and pelvic stabilizers. This effort effectively mirrored the muscle activation patterns seen in tonic–clonic seizures, combining a phase of increased muscle tone with rapid, rhythmic contractions.

The second effort (E2) was designed to closely replicate the mechanical strain characteristics observed at the muscle tissue level during tonic–clonic seizures. Since not all participants reported delayed onset muscle soreness (DOMS) after E1, this phase of the experiment focused on inducing muscle strain conditions that could lead to DOMS, as observed in patients following seizures. While E1 aimed to globally simulate the whole-body effort characteristics present in epilepsy, E2 emphasized the generation of intense mechanical tension and muscle contraction patterns associated with DOMS development. Participants first performed an isometric “wall sit” position, where their backs were fully supported against a wall, with the knees and hips bent at a 90-degree angle, and their feet providing the only point of support. This position imposed high stabilization demands, engaging both the lower limb and trunk muscles to maintain static balance. The exercise was continued until voluntary termination due to increasing discomfort and an inability to sustain the position any longer, but for no less than one minute. The sustained time under isometric tension progressively increased motor unit recruitment, leading to a more extensive and intense activation of muscle. Participants were verbally encouraged to maintain the position for as long as possible, closely simulating the intense muscle tension observed during the tonic phase of a seizure. Immediately after the inability to sustain the wall sit position, participants transitioned to performing bodyweight squats for a minimum of five minutes or until muscular exhaustion. They were instructed to perform squats with the highest possible intensity and frequency, with particular emphasis placed on tempo, reflecting the rapid, repetitive lower limb contractions characteristic of the clonic phase of a seizure. During the squat phase, the tempo of movement slowed towards the end due to muscle exhaustion. The entire group reported delayed onset muscle soreness (DOMS) in the following days. During both phases, the effort primarily engaged the lower limb muscles in an isometric contraction during the wall sit and, in the squat phase, transitioned to intense eccentric–concentric contractions, further increasing dynamic stimulation and mechanical strain. The effort in E2 primarily targeted the quadriceps, hamstrings, gluteus muscles, and the erector spinae group.

Throughout the entire effort, a physiotherapist closely monitored and supervised the participants, providing verbal cues to maintain high engagement and motivation, enforce strong mechanical tension and stimulation, and prevent technical errors.

Additionally, each participant was asked to avoid any exercise 24 h before the test and until the last blood draw.

Both sets of exercises were performed by the same group of participants with a 4-month interval draw.

### 4.2. Blood Sampling

Venous blood samples were taken from each subject three times: before the activity on the fasting state in the morning, 3 h and 24 h after the exercises. Venous whole blood samples (10 mL) were immediately centrifuged, and the serum was collected and frozen at −80 °C. All participants had their blood examined for blood count and CRP with standard laboratory tests.

### 4.3. Biomarkers Evaluation

Serum levels of MMP-9 and S100B were measured using sandwich-type ELISA in accordance with the manufacturer’s instructions. For MMP-9, we used a kit from R&D Systems (Cat no. DMP900, Minneapolis, MN, USA), and for S100B from Merck Millipore (Cat. No #EXHS100B-33K, Darmstadt, Germany). The absorbance at 450 nm was measured using a Multiscan Go spectrophotometer (Thermo Scientific, Waltham, MA, USA). The protein concentrations were calculated according to the manufacturer’s instructions.

### 4.4. Statistical Analysis

Statistica for Windows (version 13.3) was used to analyze the data and compare the groups. Results were considered significant when the *p*-value was less than 0.05 (*p* < 0.05). A one-way analysis of variance for repeated measurements was conducted to evaluate biomarker concentration levels. The data showed a normal distribution despite the small sample size. The Newman–Keuls test was employed for individual post hoc comparisons. The Student’s *t*-test was used to compare the E1 and E2 groups. The Spearman correlation test was applied to assess the relationship between age and biomarker levels at different time points.

## 5. Conclusions

Our study demonstrates that a single, brief bout of physical exercise, mimicking the exertion of a motor seizure, increases serum MMP-9 levels but does not affect S100B. We employed two distinct exercise protocols—one engaging many muscle groups simultaneously, and another targeting only selected lower limb muscles—and found similar MMP-9 increases regardless of the protocol. This pattern of MMP-9 elevation, with no accompanying increase in S100B, differs from that observed after bilateral motor seizures [4,5,8], suggesting that while physical exertion may contribute to increased MMP-9, additional mechanisms, such as blood-brain barrier disturbance, likely contribute to the elevation of both MMP-9 and S100B during seizures.

The absence of an S100B increase following exercise supports its potential as a specific biomarker for epileptic seizures, differentiating them from events that might mimic them, such as psychogenic non-epileptic seizures. However, further research is needed to confirm this specificity.

This study provides evidence that muscle contractions alone can influence serum MMP-9 levels, but the distinct patterns observed suggest that factors beyond muscle exertion contribute to the biomarker profile seen after seizures. Further research is needed to fully elucidate the mechanisms underlying biomarker release in epilepsy and to determine the specific contributions of muscle contractions versus other seizure-related processes.

## Figures and Tables

**Figure 1 ijms-26-01906-f001:**
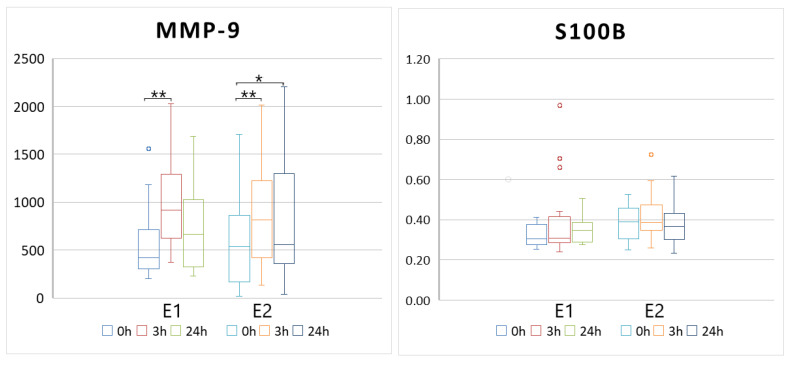
The levels of MMP-9 and S100B during both E1 and E2 physical efforts. The boxes show mean values with standard deviations and median. * *p* < 0.05 and ** *p* < 0.005 significant differences compared the time point 0 h in the appropriate group (Neuman–Keuls test).

**Table 1 ijms-26-01906-t001:** The levels of MMP-9 and S100B after the two sets of different efforts in healthy individuals. p ^1^ significant differences compared the time point 0 h in the appropriate group (Neuman–Keul test); p ^2^ significant difference between E1 and E2 biomarkers levels at the same time points (Student *t*-test).

Participants	No (Male/Female)	Mean	SD	p ^1^	p ^2^
Age	16 (4/12)	32.37	9.23		
Physical effort 1					
MMP-9 ng/mL	14 (4/10)				
0 h		583.45	385.26	**-**	0.83
3 h		999.84	536.20	0.0021	0.78
24 h		764.85	495.06	0.1070	0.12
S100B pg/mL					
0 h		32.28	5.35	-	0.025
3 h		40.86	21.58	0.16	0.78
24 h		35.27	6.87	0.21	0.47
Physical effort 2					
MMP-9 ng/mL	16 (4/12)				
0 h		603.33	478.07	-	0.83
3 h		900.54	568.75	0.006	0.78
24 h		796.87	639.01	0.031	0.86
S100B pg/mL					
0 h		38.06	9.05	-	0.025
3 h		41.80	12.39	0.57	0.78
24 h		37.36	10.59	0.97	0.47

## Data Availability

Data are available from the author of the publication on request; email: jsielczak@ipin.edu.pl.

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
