# Peer review of "A Single Bout of Physical Exercise Mimicking a Motor Seizure Increases Serum MMP-9, but Not S100B"

_ijms, 2025, doi:10.3390/ijms26051906_

Round 1

Reviewer 1 Report (New Reviewer)

Comments and Suggestions for Authors

The manuscript by Sielczak and co-authors evaluates the effect of physical exercise mimicking muscle activity during bilateral tonic-clonic seizures on the serum levels of MMP-9 and S100B, which are two biomarkers associated with blood-brain barrier integrity. The study explores an interesting topic and presents a potentially valuable contribution to the field, but several issues reduce its impact. Below there is a list of my specific concerns and criticism:

1. The novelty of the manuscript is limited. Similar studies exploring the influence of exercise on serum biomarkers (especially MMP9) have been conducted and published. The specific approach of mimicking seizure-induced muscle activity adds an interesting angle, but the findings mostly confirm existing knowledge that MMP-9 levels increase with physical effort, while S100B remains stable. The authors did ot explore the observation mechanistically, which diminishes the significance of the paper.

2. The results are not entirely supported by the data. The sample size is small (16 participants), which limits the statistical power and generalizability of the findings. In addition, the lack of detailed individual data and the reliance on a few time points for biomarker analysis weakens the conclusions. Also the figures do not add significant value or new understanding.

3. The manuscript would benefit from better controls. For example, including a comparison to patients with actual tonic-clonic seizures would help differentiate exercise-induced biomarker changes from those caused by epileptic activity. Furthermore, assessing additional biomarkers would enhance the study. 

In summary, the manuscript provides a limited contribution to the understanding of exercise-induced changes in serum biomarkers relevant to epilepsy. The authors did very few experiments and made only limited numer of observations. I would recommend this manuscript to be rejected. 

Comments on the Quality of English Language

The quality of presentation requires improvement. The writing is repetitive and contains grammatical errors. Additionally, the figures are basic and their quality could be improved. Including more sophisticated statistical analyses and comprehensive data visualization would strengthen the manuscript.

Author Response

This study's novelty lies in its rigorous approach to measuring MMP9 and S100B in a carefully selected group of healthy adults, excluding numerous conditions that could confound these measurements. The standardized exercise protocol, performed by all participants in a controlled environment, further strengthens the study's design. While the sample size of 16 healthy adults is relatively small, it is sufficient to detect meaningful differences in biomarker levels.

We acknowledge that the primary aim of this work was to investigate whether a brief, seizure-like motor effort is sufficient to induce changes in these biomarkers, and to determine the duration of any such changes. The study is simple and statistical analysis is appropriate to show the changes between groups. It is important to note that while numerous studies have measured MMP9 and S100B in epilepsy and other conditions, many fail to account for confounding factors that can influence these biomarkers. Our study specifically addresses the impact of physical exertion, a variable often overlooked. We previously observed elevated MMP9 and S100B levels following motor seizures in patients, which directly motivated the present experiment. This prior work is discussed within the paper. Furthermore, we emphasize the critical importance of the time interval between exercise and biomarker measurement. Without careful consideration of this timing, data can become noisy and lead to seemingly contradictory results. This principle is further illustrated for example in our published work on epilepsy and BDNF, where we demonstrated that the timing of BDNF measurement relative to the last seizure significantly affects the observed levels. Specifically, we found that BDNF levels in epilepsy patients deviate from controls after seizures and the ‘stable’ level is the same as in age and sex matched control when we kept at least 7 days from seizures.

In response to the reviewer's feedback, we have shortened the manuscript, improved the English language, and clarified the study's focus. Our primary objective is not to explain the presence or absence of MMP9/S100B elevation, but rather to demonstrate the effect of brief physical exertion on the levels of these biomarkers.

Additionally, in response to the other reviewer's concerns, we have significantly expanded and refined the description of the exercise protocols used in our study. The revised manuscript now provides a much more detailed characterization of the physical efforts, specifying the exact movements, their intensity, and their physiological rationale. We have elaborated on the targeted muscle groups and the mechanical stresses induced by each phase of the exercises, particularly in relation to the muscle activation patterns observed during tonic-clonic seizures. Furthermore, we have clarified the modifications available to participants who were unable to sustain the initial exercise intensity and ensured that the methods accurately reflect the effort required to simulate seizure-like muscle contractions. This enhanced description strengthens the study's methodological transparency and improves its reproducibility.

I hope that you find this explanation accurate.

Reviewer 2 Report (New Reviewer)

Comments and Suggestions for Authors

The manuscript that entitled by “A Single Physical Exercise Imitating Muscle Activity During an Epileptic Seizure Causes an Increase of MMP-9 Serum Level but Does Not Increase S100B”, conducted a research study to the aim of the study was to evaluate two widely studied biomarkers: MMP-9 and S100B in blood serum in healthy people after physical exercise mimicking physical activity during bilateral tonic-clonic seizures. Their findings stated that both sets of motor exercises led to a similar increase in MMP-9 levels, while neither affected S100B levels. The manuscript is generally well-addressed and well-cited; However, I have some comments and suggestions.

Line 4: The title is too long. I suggest changing it to be more focused and specific such as: Levels of MMP-9 and S100B  biomarkers during An Epileptic Seizure at motor exercises.

Line9: The abstract started by the aim of study. The abstract preferred to start by the background first by one short sentence first then aim of study.

Line 25: I suggest adding your main used biomarkers in your study; MMP-9 9 and S100B

Line 87: At the end of introduction please,  briefly mention and highlight the main conclusions or your findings.

Line 89: There is a lot of data missing in the section of materials and methods, then some of them mentioned here at results section which confused me more. Therefore, I suggest moving the results to be AFTER materials and method section. That's will have good consequences for your study.

Line 114: I suggest getting the chart colored for better presentation and easily comprehensible for readers.

Line 231: Please add the approval number and year from ethical committee.

Line 234: Please mention the number of participants shared in this conducted study with more details about age and sex, body  if available.

Line 239: There is a lot of data concerning the participants and I wish to add a table showing their data in detail and divide the columns of table according to the group type.

Line 253: I suggest adding drawing colored cycling diagram showing the previous steps with a positions, that will be more interesting and deliver your idea by easy way.

Line 254: Please mention the area or names of targeted muscles for this contraction in E2 effort.

Line 259: What kind of squats are they used? Which area of body targeted? Please mention in detail.

Line 262: Please explain why you chose 4 months interval time to repeat the exercises. Also, why didn’t you take 3 times to get more data then get more accurate average for these data readings.

Line 266: How long fasting state they did?

Line 266: Please explain why you chose this time frame for collection of sample. Why is it 3 hours after exercise? It can be 6 or 9 or 12 hours after exercise. Please show the significance of your plan to do that.

Line 273: I hope you add the catalog number for those both kits.

Line 287: The conclusion is long, please revise to be more focused. It should summarize the key findings of your study, reiterate the main points of your argument, highlight the significance of your research, and discuss potential implications and future research directions while avoiding introducing new information.

References number 19 is incomplete. Upon journal guidelines for journal articles, it should have Abbreviated Journal Name YearVolume and page range. Please revise throughout the manuscript.

Comments on the Quality of English Language

Major editing of English language required.

Author Response

Answers to Reviewer 2.

We are grateful for remarks and question of our work. Please find below our answers and explanations.

Line 4: The title is too long. I suggest changing it to be more focused and specific such

as: Levels of MMP-9 and S100B  biomarkers during An Epileptic Seizure

at motor exercises.

We have changed title – now it is shorter and more to the point.

Line9: The

abstract started by the aim of study. The abstract preferred to start by the

background first by one short sentence first then aim of study.

We changed the abstract to start it from the background.  

Line 25: I suggest adding your main used biomarkers in your study; MMP-9 and S100B

We add them to keywords

Line 87: At the end of introduction please,  briefly mention and highlight the

main conclusions or your findings.

We added main conclusion as the last sentence to the Introduction part.

Line 89:

There is a lot of data missing in the section of materials and methods, then

some of them mentioned here at results section which confused me more.

Therefore, I suggest moving the results to be AFTER materials and method

section. That's will have good consequences for your study.

I think that this is because of the schedule of journal used. But we may change it if necessary. We describe more precisely the experimental group and the sets of exercises to explain the experiment. 

Line 114: I suggest getting the chart colored for better presentation and easily

comprehensible for readers.

We added color figure 1

Line 231:

Please add the approval number and year from ethical committee.

We added the number of the agreement of ethical committee (IPIN LBC No 18/2021).

Line 234: Please mention the number of participants shared in this conducted study with more details about age and sex, body  if available.  

W added in the Methods part the more precise description of the participants as it was mentioned above.

Line 239:

There is a lot of data concerning the participants and I wish to add a table showing their data in detail and divide the columns of table according to the group type.

We conducted a medical interview and excluded participants based on the contraindications we proposed. We also gathered more data to ensure that all volunteers were free of chronic diseases such as hypertension, hypercholesterolemia, or obesity. However, we did not collect data on their weight or blood pressure. The volunteers did not take any medications chronically, and we excluded individuals with acute diseases prior to the exercise, as well as those with additional issues affecting their ability to engage in physical activity.

Line 253: I suggest adding drawing colored cycling diagram showing the previous steps with a positions, that will be more interesting and deliver your idea by easy way.

A color drawing illustrating the experimental setup is included as an attachment.

Line 254: We have supplemented the work with a much more detailed description of the performed efforts and provided an explanation of the muscle engagement.

Line 259: We have described the type of squats performed as classic bodyweight squats, executed within the maximum tolerated range of motion.

Line 262: Please explain why you chose 4 months interval time to repeat the exercises. Also, why didn’t you take 3 times to get more data then get more accurate average for these data readings.

The time interval between exercise sessions should be at least 7-14 days, as our previous studies in epilepsy have shown that biomarker elevations can persist for up to 7 days. Since participants experienced no muscle weakness or pain after the first exercise session, we designed a second exercise protocol to induce such symptoms. However, logistical constraints, including scheduling and the same group of participant recruitment for multiple blood draws, necessitated a 4-month interval between sessions. We know that more blood draw would give us more information but the same schedule we had for our epilepsy patients after seizures. We just kept similar protocol.

Line 266: How long fasting state they did?

About 12 hours from the supper.

The all exercises were done in the morning so the first blood draw was in the morning about 8 o’clock for every participant, next at 11 o’clock and next in the next morning.

Line 266:

Please explain why you chose this time frame for collection of sample. Why is it 3 hours after exercise? It can be 6 or 9 or 12 hours after exercise. Please show the significance of your plan to do that.

We chose the time frame from our previous studies with patients after seizures. There ( paper by Cudna et al 2024) we collected blood 1-3 hours after seizures 24 hours and 72 hours after seizures. Other researchers collected blood 6 hours after seizures and 24 hours after seizures.

We would like to make the similar time points to the epilepsy studies.

Line 273: I hope you add the catalog number for those both kits. We added.

The calalog numbers of the kits:

MMP9 - DMP900

S100B - #EXHS100B-33K

Line 287: The conclusion is long, please revise to be more focused. It should summarize the key findings of your study, reiterate the main points of your argument, highlight the significance of your research, and discuss potential implications and future research directions while avoiding introducing new information.

We shortened the conclusion parts removing all the hypotheses and additional remarks.

References number 19 is incomplete. Upon journal guidelines for journal articles, it should have Abbreviated Journal Name YearVolume and page range. Please revise throughout the manuscript. We have changed it according to the rules.

Round 2

Reviewer 1 Report (New Reviewer)

Comments and Suggestions for Authors

The authors have made some improvements in data presentation, methodology description, and results discussion. However, no new experiments related to the original submission were conducted. Considering the nature of the study, the reviewer does not support the publication of this manuscript in IJMS. 

Comments on the Quality of English Language

None.

Author Response

Dear Reviewer,

We sincerely appreciate your time and effort in reviewing our manuscript and providing valuable feedback. While we understand your concerns, we believe that our study offers novel insights into the dynamics of biomarkers in response to diffrent types of physical exertion. In response to the review process, we have introduced additional refinements to the discussion section to further clarify our findings.

We acknowledge the limitations of our study; however, we ensured that our study group was homogeneous, consisting of healthy individuals without comorbidities, and that the applied exercise protocol was standardized and repeatable. This experimental approach provides valuable insights into how even very short exertion can influence biomarker concentrations. Our findings suggest that future biomarker studies should carefully monitor participants' physical activity, even intensive activities of daily living, as even short exertion may impact the results.

It is also important to note that very few studies have examined how even short, intense exertion may impact biomarkers. We believe our approach adds a unique perspective to the existing literature. We would be grateful for your reconsideration of our manuscript and your evaluation of the revised version.

Best regards,
Jan Sielczak

Reviewer 2 Report (New Reviewer)

Comments and Suggestions for Authors

The manuscript is greatly improved!. Thank you for taking my comments into your consideration. 

Comments on the Quality of English Language

Minor English language editing required.

Author Response

The manuscript is greatly improved!. Thank you for taking my comments into your consideration. 

Dear Reviewer,

Thank you for your constructive feedback and for taking the time to reconsider our manuscript. We appreciate your acknowledgment of the improvements made in response to your previous comments. We believe the revisions have enhanced the clarity and quality of the manuscript. Thanks to your suggestions, the manuscript has become stronger and has gained in scientific quality.

We look forward to your final evaluation and hope the revised version meets your expectations.

Best regards,
Jan Sielczak

This manuscript is a resubmission of an earlier submission. The following is a list of the peer review reports and author responses from that submission.

Round 1

Reviewer 1 Report

Comments and Suggestions for Authors

Sielczak et al. present an interesting work showing that a single physical exercise can replicate increase of MMP-9 in epileptic convulsive seizures but not S100B which were thought to be the markers of blood brain barrier (BBB) leakage following bilateral tonic-clonic seizures in humans. The current study revealed that increased MMP-9 can be confounded by extensive muscle activity not necessarily a result of BBB disruption in convulsive seizures. The study cautions the interpretation of elevated MMP-9 in epilepsy as biomarker of BBB permeability as it levels increase in strenuous exercise. However, excessive muscle contractions do not appear to affect the levels of S100B, suggesting that upregulation of S100B may be specific to BBB disruption following tonic-clonic seizures. Therefore, S100B could be used as a potential biomarker in predicting the degree of BBB leakage in epilepsy patients. As I am convinced of the significance of this work, the lack of proper experimentations and assumptions/claims made from extremely limited data in the manuscript further weaken the study. Therefore, I have several concerns that need to be addressed to improve the quality of the manuscript.

Major Comments

Positive controls for each analyte from epileptic patients should be included in the analysis for comparison. This will help determine the extent to which baseline levels of MMP-9 and S100B differ between epileptic patients and healthy individuals. Additionally, it would be valuable to know if the increased levels of MMP-9 at 3 hours and 24 hours following exercises 1 and 2 are comparable to those observed in epileptic patients with bilateral tonic-clonic seizures at the corresponding time points.

How were the levels of these markers in non-convulsive seizures or status epilepticus? Are they also elevated? Are they different from healthy individuals? This needs to be also discussed in the manuscript.

Was the normality of the data tested? If so, What type of test was used? Including asterisks on the graphs would be helpful to indicate significant differences between groups. Tittle for Y-axes are missing. What do the bar graphs and error bars represent? All these should be clarified in the manuscript/legends.

Minor comments

There are typos throughout the manuscript. Please review and correct them.

The catalog numbers or RRIDs of the kits used in the experiments should be listed.

Author Response

Dear Sir,

We are grateful for the reviewers comments and improvement. Please find below the answer to reviewers comment.

To Reviewer  1

In response to the reviewer’s suggestions, we have made changes to the manuscript by referring to our previous studies in which the release of biomarkers was assessed in other situations, allowing us to relate these data to our current work. This enabled us to compare baseline levels of MMP-9 and S100B between patients with epilepsy and healthy individuals.

Regarding non-convulsive seizures and status epilepticus, we did not find such studies, and this is also mention in the manuscript.  Indeed, there is also a lack of data regarding MMP-9 levels in healthy individuals who have not undergone any intervention. We suggest that including such a control group in future studies would be valuable in better understanding the natural fluctuations in this biomarker.

We discuss these problems in manuscript. (the last paragraph of the discussion):

The patients with epilepsy have they basic level of MMP-9 and S100B higher than the age and sex matched control without epilepsy and other neurological diseases. [3] Specifically, the mean (±SEM) serum concentration of MMP-9 in the epilepsy group was 846.66 ± 56.35 ng/ml, compared to 533.35 ± 32.89 ng/ml in the control group, aligning with the levels observed in the current study at time point 0. Similarly, the concentration of S100B was significantly elevated in the epilepsy group, while the con-trol group had S100B levels comparable to those in our current study at time point 0 (23.63 ± 3.13 pg/ml vs. 32.28 ± 5.35 pg/ml). After seizures, the concentrations of MMP-9 and S100B increase for at least 24 hours, and MMP-9 is elevated for 72 hours [4,5,8]. In our current study, we observed a 50% increase in MMP-9 concentration in healthy volunteers following an exercise session, with no corresponding increase in S100B. Nass et al. [5] reported that MMP-9 levels increased by approximately 50% and S100B by 80% after seizures. Cudna et al. [4,8] found that MMP-9 and S100B concentrations were approximately 100% higher than control levels 3 hours post-seizure. However, since the baseline biomarker levels in the epilepsy group were not reported in that study, a direct comparison to our findings is challenging. In conclusion, physical activity induces a smaller and shorter increase in MMP-9 concentration compared to seizures and does not elevate S100B levels. This suggests that S100B might be a more specific biomarker of seizures than MMP-9 which are more dependent on motor activity during tonic-clonic muscle contractions. Investigating non-motor seizures could help clarify the extent to which the increase in serum MMP-9 levels is influenced by muscle activity versus brain activity itself. The majority of studies examining MMP-9 and S100B serum concentrations following seizures focus on generalized tonic-clonic seizures or febrile seizures, with little data available on non-motor seizures or status epilepticus. Recent meta-analyses of MMP-9 serum concentrations have also reported consistent findings across studies [30].

The concentration of certain biomarkers has been suggested as a potential predictor of epileptic seizures [9]. As highlighted in our previous studies, serum levels of MMP-2, MMP-9, and CCL-2, as well as other markers related to blood-brain barrier (BBB) activation, were found to predict seizure frequency over a 12-month period. These biomarkers may indicate underlying processes such as BBB activation, neuronal excitability, and inflammation, which are involved in seizure initiation.

We think that this discussion is enough comprehensive, and we cannot compare the data from the epilepsy studies because of different methodology and no basic level for epilepsy patients group.

  1. We made minor improvements to both figures, added asterisks where necessary, and enhanced the descriptions to ensure clarity. We apologize for the mistakes we made.

  2. We reviewed the document for English spelling and grammatical issues and made corrections to improve accuracy.

Reviewer 2 Report

Comments and Suggestions for Authors

In this research article, the authors examine two biomarkers of epilepsy after seizure-imitating physical exercises. They measure in 12 healthy persons the serum levels of MMP-9 and S100B. They look for relationships between exercise and protein levels, and claim, thatS100B is not changing, so it may be used as a biomarker of epilepsy contrary to MMP-9.

The number of participants is low, and in the case of MMP-9 level the SD of the results is very high. Even if they may find statistical correlation, this experimental setup and the numerical results are not convincing. In the literature different results are found in the case of both parameters concerning physical activities. For example:  in Exp Physiol 90.4 pp 613–619 it is said that MMP-9 level is not changing in rats following different levels of muscle activity. In case of S100B the authors admit, that the level of the protein in the serum may drop in short interval. These proteins may be markers of epilepsy without seizures, if I understand correctly.

In summary: these experiments with just a few participants, and the questionable statistical analysis are not providing convincing results to use any of the markers in medical practice.

Author Response

Thank you for the reviewer’s comments. We appreciate the feedback and would like to address the concerns raised regarding the number of participants, statistical analysis, and the applicability of the biomarkers.

Firstly, we acknowledge that the sample size in our study is limited, and we agree that increasing the number of participants could strengthen the statistical power of the results. However, this study was conducted in a controlled environment (healthy subjects, the same participants for both sets of exercises, the controlled and supervised exercise sessions) to explore the potential relationship between exercise and the levels of MMP-9 and S100B,  and we believe the results provide valuable results.

We have revised and checked the statistical analysis and made some minor improvements to ensure that the presented data and analysis are correct. The presentation in figures and tables is better described. We have checked the correctness of our analysis and we want to emphasize that our small sample for all groups meets the criteria for a normal distribution of data. This allowed us to use Anova for repeated measures and as a post-hoc analysis the Neumann-Keuls test. This is a correct approach, although conservative and simple, but sufficient to obtain correct results. As for the high standard deviation observed for MMP-9 levels, we recognize this variability and it may be due to the different age and training of the study participants.

We would like to point out that there are few studies similar to ours that examine isolated physical exercise designed to simulate seizure-like activity. Previous research has typically involved more complex activities such as long-distance swimming, running, or other less controlled forms of exercise. These studies often focus on a broader range of activities in less regulated environments. In contrast, our study specifically investigated a very short, intense exercise, which makes it more comparable to the conditions that occur during an epileptic seizure. This distinction is important as it allows for a clearer understanding of how exercise may influence biomarker levels in a manner more closely related to seizure events. We believe this methodological approach provides a novel perspective and could serve as a foundation for future studies exploring the link between physical exertion and epilepsy.

Regarding S100B, we have clarified in the manuscript that while we did not observe significant changes in serum levels of this protein post-exercise, there is evidence in the literature suggesting that S100B levels can fluctuate over shorter time intervals. Our findings support the idea that S100B may be a more stable biomarker compared to MMP-9, especially in the context of epilepsy, but further research is needed to validate its role as a reliable biomarker in non-seizure conditions.

In conclusion, while we agree that further studies with larger sample sizes and refined statistical analyses are necessary to validate these findings, we believe our study contributes to the understanding of the dynamics of MMP-9 and S100B in response to physical exercise and their potential role as biomarkers of epilepsy. We are confident that our study provides a foundation for future research in this area.

Thank you for the reviewer’s comments. We appreciate the feedback and would like to address the concerns raised regarding the number of participants, statistical analysis, and the applicability of the biomarkers.

Firstly, we acknowledge that the sample size in our study is limited, and we agree that increasing the number of participants could strengthen the statistical power of the results. However, this study was conducted in a controlled environment (healthy subjects, the same participants for both sets of exercises, the controlled and supervised exercise sessions) to explore the potential relationship between exercise and the levels of MMP-9 and S100B,  and we believe the results provide valuable results.

We have revised and checked the statistical analysis and made some minor improvements to ensure that the presented data and analysis are correct. The presentation in figures and tables is better described. We have checked the correctness of our analysis and we want to emphasize that our small sample for all groups meets the criteria for a normal distribution of data. This allowed us to use Anova for repeated measures and as a post-hoc analysis the Neumann-Keuls test. This is a correct approach, although conservative and simple, but sufficient to obtain correct results. As for the high standard deviation observed for MMP-9 levels, we recognize this variability and it may be due to the different age and training of the study participants.

We would like to point out that there are few studies similar to ours that examine isolated physical exercise designed to simulate seizure-like activity. Previous research has typically involved more complex activities such as long-distance swimming, running, or other less controlled forms of exercise. These studies often focus on a broader range of activities in less regulated environments. In contrast, our study specifically investigated a very short, intense exercise, which makes it more comparable to the conditions that occur during an epileptic seizure. This distinction is important as it allows for a clearer understanding of how exercise may influence biomarker levels in a manner more closely related to seizure events. We believe this methodological approach provides a novel perspective and could serve as a foundation for future studies exploring the link between physical exertion and epilepsy.

Regarding S100B, we have clarified in the manuscript that while we did not observe significant changes in serum levels of this protein post-exercise, there is evidence in the literature suggesting that S100B levels can fluctuate over shorter time intervals. Our findings support the idea that S100B may be a more stable biomarker compared to MMP-9, especially in the context of epilepsy, but further research is needed to validate its role as a reliable biomarker in non-seizure conditions.

In conclusion, while we agree that further studies with larger sample sizes and refined statistical analyses are necessary to validate these findings, we believe our study contributes to the understanding of the dynamics of MMP-9 and S100B in response to physical exercise and their potential role as biomarkers of epilepsy. We are confident that our study provides a foundation for future research in this area.

Round 2

Reviewer 1 Report

Comments and Suggestions for Authors

Thank you to the authors who have sufficiently addressed my comments. The current version of the manuscript is suitable for publication.

Reviewer 2 Report

Comments and Suggestions for Authors

I have read the answers of the authors and the modified version of the manuscript. I still have the feeling that working with such large SD is not safe, even though there are statistical methods to compare these numbers. In addition, there are 14 samples at exercise 1, and 16 at the 2nd occasion, which is not explained (or at least I could not find it).

I do not understand the main point of the paper: if the authors try to imitate seizure-like muscle activities, than why they think, that which molecule level is not changing is the better marker in case of epilepsy. Then the simulation is not correct. If it is already known that the basic level of the examined proteins are elevated in epileptic patients, why do we look at the level of them after seizure?